# Standardized Methodology for Target Surveillance against African Swine Fever

**DOI:** 10.3390/vaccines8040723

**Published:** 2020-12-02

**Authors:** Stefano Cappai, Sandro Rolesu, Francesco Feliziani, Pietro Desini, Vittorio Guberti, Federica Loi

**Affiliations:** 1OEVR—Sardinian Regional Veterinary Epidemiological Observatory, Istituto Zooprofilattico Sperimentale della Sardegna “G. Pegreffi”, 09125 Cagliari, Italy; stefano.cappai@izs-sardegna.it (S.C.); sandro.rolesu@izs-sardegna.it (S.R.); 2Italian Reference Laboratory for Pestivirus and Asfivirus, Istituto Zooprofilattico Sperimentale dell’Umbria e delle Marche “Togo Rosati”, 06126 Perugia, Italy; f.feliziani@izsum.it; 3ATS Sardegna, ASSL Sassari, Servizio di Sanità Animale, 07100 Sassari, Italy; pietro.desini@atssardegna.it; 4ISPRA—Institute for Environmental Protection and Research, 00144 Roma, Italy; vittorio.guberti@isprambiente.it

**Keywords:** African swine fever, passive surveillance, wild boar, GIS-technology, Sardinia, disease eradication, exit strategy

## Abstract

African swine fever (ASF) remains the most serious pig infectious disease, and its persistence in domestic pigs and wild boar (WB) is a threat for the global industry. The surveillance of WB plays a central role in controlling the disease and rapidly detecting new cases. As we are close to eradicating ASF, the need to find any possible pockets of infection is even more important. In this context, passive surveillance is the method of choice for effective surveillance in WB. Considering the time and economic resources related to passive surveillance, to prioritize these activities, we developed a standardized methodology able to identify areas where WB surveillance should be focused on. Using GIS-technology, we divided a specific Sardinian infected area into 1 km^2^ grids (a total of 3953 grids). Variables related to WB density, ASF cases during the last three years, sex and age of animals, and the type of land were associated with each grid. Epidemiological models were used to identify the areas with both a lack of information and an high risk of hidden ASFV persistence. The results led to the creation of a graphic tool providing specific indications about areas where surveillance should be a priority.

## 1. Introduction

African swine fever (ASF) is a devastating disease for both domestic and wild pig populations, representing the primary challenge for the whole European agricultural sector, in particular the pig farming industry [1]. ASF disease is caused by the African swine fever virus (ASFV), a large, enveloped double-stranded DNA virus, the only member of the Asfarviridae family [2], which mainly infect myeloid cells, such as monocytes, macrophages, and dendritic cells [3].

The negative consequences of ASFV spread are of global proportion [4,5,6]. The economic and social consequences amount to dozens of billions of euros lost yearly and potential repercussions for individual livelihoods and national food security [7,8,9]. The disease’s epidemiological circle does not have a unique identification and includes not only domestic pigs (*Sus scrofa domesticus*) and wild species (e.g., wild boar), as reported in Europe, but also *Phacoerus africanus* or *Potamochoerus* spp. in Africa, with complications due to the presence of soft ticks as vector (genus *Ornithodoros*) [10,11,12,13].

Given ASF’s remarkable capacity for transboundary and transcontinental spread [14], its average speed of propagation of 250 km/year in Europe [15] (which is even faster in Asia (550 km/year) [16]), and in the absence of a licensed vaccine or treatment, it is not surprising that from 2014 to date, the disease has affected 55 countries worldwide [17].

Considering the different target populations that can be affected by ASF, the disease’s eradication can be declared only if no outbreaks occur in domestic pigs or if cases in wild boar are no longer detected. Even under the condition of no virus being detected in both populations but the ongoing presence of seropositive animals, the affected countries will have to retain rigorous control measures to control/eradicate the disease [5,18,19,20,21,22]. Although the disease management measures established by the Council Directive 2002/60/EC provide the tools for ASFV control in the EU, the lack of a licensed vaccine and the existence of knowledge gaps in several critical areas of ASF epidemiology are impediments that need to be addressed.

The working document 7113/2015 by the European Commission underlines the importance of passive surveillance in affected areas without ASF cases in wild boar from 4 to 5 months (particularly when the full summer period is included). The main measures to be applied include an active search for wild boar carcasses by trained staff and a total ban on feeding [23]. The fundamental role of passive surveillance in ASFV detection has been underlined by several studies [1,24,25,26]. Furthermore, as recently demonstrated by Gervasi in 2020 [27], although a decrease in ASF prevalence corresponds to a decrease in the probability to find the virus, searching for carcasses is the most useful method to find the last ASFV cases. On the other hand, to be efficient, passive surveillance should be based not only on samples collected from wild boar killed in road traffic accidents [28] but also on enhanced surveillance based on the active searching of carcasses, preferably in small areas where the presence of residual viral pockets cannot be excluded [27].

Standardized, well-planned passive surveillance is necessary in European countries (i.e., parts of Latvia, Estonia, and Italy), previously declared to be ASF-endemic countries, where the disease persists only in seropositive wild boar with low prevalence. In such contexts, recreational hunting is not able to provide adequate samples to detect possible residual pockets of infection [22]. The lack of well-planned surveillance and, more generally, the lack of ongoing ASF control measures (i.e., active and passive surveillance strategies and conventional biosafety and sanitary measures) hinder final eradication [29].

Despite the evident ASF decreases in Latvia during the last five years, in 2020, Oļševskis underlined the importance of maintaining effective hunting and surveillance, combined with the efficient detection and removal of wild boar carcasses, as long as ASF is present [21]. Furthermore, the authors promoted the need to implement such measures not only when ASF is present but also before its arrival [26]. In Estonia, after more than 1 year of epidemiological silence (from February 2019), the detection of a new virus is complicating eradication. The main hypothesis is the recrudescence of the virus is caused by its new introduction. Nevertheless, uncertainty is associated with the decline in the number of wild boars sampled through passive surveillance during the last five years [24]. The importance of passive surveillance in early ASF detection was demonstrated during the recent disease incursion in Brandenburg (Germany) [30].

In Sardinia (Italy), the available data from active surveillance suggest a clear decline in ASFV, leading to its complete disappearance. However, the limited samples from passive surveillance and the disease’s seasonality hinder a definitive answer for disease eradication [22]. Thus, the poor data from passive surveillance generate uncertainty in the ASF eradication process. In this epidemiological landscape, the need to maintain an high level of wild boar surveillance after the last virus identification is essential. A standardized approach able to modulate wild boar surveillance with the available epidemiological data is fundamental for program-specific control measures aimed at minimizing the risk of undetected ASFV at a very low prevalence. Otherwise, considering the difficulty of implementing this procedure throughout the region with the same intensity and accuracy, it is necessary to geographically identify the most susceptible areas.

To overcome this problem, the epidemiological data of the ASF epidemic area in Sardinia (Italy) during 2017–2020 were used to develop a standardized method based on epidemiological models to identify high-risk areas where passive surveillance should be improved. The merger of this spatial information with GIS-technology provides fundamental tools to quickly and accurately analyze specific passive surveillance aimed at certifying the absence of ASFV in wild boar.

## 2. Materials and Methods

### 2.1. Sardinian Epidemiological Landscape of ASF in 2019–2020

In Sardinia (Italy), the last virus notification (PCR-positive) in wild boar was observed on 4 April 2019. This notification emerged from two animals found dead during specific targeted passive surveillance within a protected forest in the Bultei municipality (Sassari province; area = 15.87 km^2^). As previously demonstrated, both the viral prevalence and seroprevalence have drastically decreased over the last five years across the wild boar infected zone (ZI) [22]. Otherwise, the main surveillance for ASF is limited to hunting activities during November–January, while the active searching of carcasses is almost absent, and passive surveillance is limited to wild boar roadkill [22]. More precisely, from the last PCR-positive detection until October 2020, a total of 10,869 wild boar were tested for ASF: 95% (10,360) were observed through hunting, while 5% (509) were found via passive surveillance. Most of the passive surveillance samples (56%) were concentrated during the same months as the hunting season (November–January). As shown in Figure 1, the current epidemiological context indicates the complete absence of ASFV and the presence of seropositive animals limited to two main areas: Goceano–Baronia (3953 km^2^) and Barbagia–Ogliastra (1896 km^2^).

### 2.2. Data Collection and Management

Detailed ad hoc databases were created using Microsoft Access to collect information from the internal database systems (SIGLA) of the Istituto Zooprofilattico Sperimentale della Sardegna (IZS-Sardegna). Data about hunted wild boar from the two areas of Goceano–Baronia and Barbagia–Ogliastra between November 2017 and January 2020 (three hunting seasons) were collected. More precisely, data on the sample (yy/mm/dd), province, municipality, forest location, latitude and longitude, type of sample (organs, tissue, serum, or blood), sex and age of the wild boar, and laboratory results (virological and serological tests) were included. Based on latitude and longitude, features were associated with quadratic grids of 1 km^2^.

The presence of young (0–18 months) or adult (>18 months) PCR-positive or seropositive animals, animal density estimation [13], number of tested wild boar and, consequently, the compliance value (as noted below) were determined for each grid and by year. Furthermore, given the dynamic nature of wild boar, the same values were imputed in the neighboring grids considering the average distance as 5 km^2^ for the radius [31]. In addition, the altitude (mamsl), road (km), amount of forest (km^2^), and protected forest (km^2^) were collected by Corine Land Cover 2012 (https://land.copernicus.eu/pan-european/corine-land-cover/clc-2012/view). This procedure was realized using ArcGIS^®^ (ArcMap software by Esri, version 10.4, Environmental Systems Research Institute: Redlands, CA, USA) geoprocessing tools to join all layers built for each variable with different options. A geodesic buffer was created with the standard planar method to consider the joined attributes related to the wild boar’s home range, and all buffers were dissolved together into a single feature, thus removing any overlap. In addition, the geometric intersection of the input features was used for the water, streets, forest, and protected areas, and all attributes from the input features were transferred to the output feature class. This provides the sum of the variable values in each point or buffer. Figure 2 illustrates the flow-chart of the study design.

### 2.3. Statistical Analysis

After the appropriate data checks, descriptive analyses were carried out. Common synthesis measures were used as the mean (standard deviation (SD)) or median (I–III interquartile (IQR)) for the quantitative variables, whereas the qualitative variables were summarized as the frequency and percentage. To compare the qualitative variables, either a Chi-square test or a Fisher’s exact test was applied. Considering the baseline distribution, differences between quantitative variables were evaluated by applying Student’s T-test or the Kruskal–Wallis nonparametric test.

As in our previously published risk analyses, the main risk factors indicating the presence of undetected pockets of infection were the recent detection of PCR-positive animals, the presence of young seropositive animals, and the lack of information [13,32]. Two models were used to understand what variables primarily describe the presence of PCR-positive animals (outcome 1) or the presence of young seropositive animals (outcome 2), while maintaining the same environmental variables. To provide more recent results, given the complete absence of PCR-positivity during 2019–2020 but the persistence of young seropositive animals, outcomes 1 and 2 were associated with two different time periods, namely hunting season 2018–2019 and hunting season 2019–2020, respectively.

Considered as epidemiological units, the grids were defined as positive if containing one or more infected animals (i.e., PCR-positive or young seropositive) and negative otherwise. Furthermore, as areas with a lack of information are the most commonly targeted areas for surveillance, a variable named compliance was calculated for each epidemiological unit (1 × 1 km^2^ grid). Supposing that 45% of the total wild boar population is hunted every year, considering the *WBh* as the number of wild boars hunted (and consequently tested) for ASF, and *WBd* the estimated wild boar population density [13], the compliance *C* is given as follows:(1)$$C_{t,j}=\frac{{\mathrm{WBh}}_{t,j}}{{\mathrm{WBd}}_{t,j}\;\times 0.45}\;$$
where *t* is the year in study, and *j* is a single grid. The compliance is used here to identify the areas with a number of tested animals lower than the median.

To keep the model estimations unbiased, the multicollinearity between the variables was tested by applying the Spearman non-parametric correlation coefficient and subsequently based on the variance inflation factor (VIF) [33]. If multicollinearity was detected, we removed one or more of the highly correlated predictors in the models. Assuming that the observations between the grids were not independent, we applied a mixed-effect logistic regression (MELR) model to each outcome, treating each grid as a random sample from a larger population and modelling the between-grid variability as a random effect. Thus, the model for observations *i* is
(2)$$\mathrm{logit}\;\left(P(y_{i}=1\right)=\beta_{0}+\sum_{k=1}^{m}\beta_{k}X_{j\left[i\right]}+\alpha_{j\left[i\right]}+\varepsilon_{j\left[i\right]}$$
for grids *j* = 1, …, n. The expected probability of outcome 1 or 2 for a grid *j* is described by a binomial distribution; the intercept *β*_0_ and the summitry vector of *m* covariates (X) measured at each grid *j* and their respective coefficients *β_k_* represent the fixed proportion of the model; *α_j_* is the random effect representing the dispersion among grids; and the n × 1 vector of errors *ε_ij_* is assumed to be multivariate and normal, with a mean of 0 and a variance matrix of $\sigma_{\varepsilon }^{2}ℜ$.

Data from Goceano–Baronia were used as the training dataset to fit the models and for internal validation based on different data periods, while the Barbagia–Ogliastra dataset was used as the test dataset for external validation. A forward–backward stepwise procedure was applied to the training dataset for variable selection, and the best fitting model was chosen based on the adjusted R^2^ values and the Bayesian and Akaike Information Criteria (BIC and AIC). All interactions with a supposedly biologically valid foundation were tested. The intraclass correlation coefficient (ICC, i.e., the correlation between the latent linear responses conditional on the fixed-effects covariates) was calculated for each MELR model to evaluate how strongly values in the same grids resembled each other. The two models results are presented as the adjusted odds ratio (OR_adj_) calculated by the Lemeshow and Hosmer method [34].

The predicted values were computed in both the training and test datasets for both the outcomes to carry out internal and external validation. The ability of the model to correctly distinguish between the two classes of outcomes (positive/negative) was first evaluated in the contingency table, and the sensitivity and specificity were calculated [35]. Furthermore, the performance of the models was measured by the C statistic and reported as the full value, median, half of the interquartile range (QRNG/2), and ratio between QRNG/2 and the median (C Var) for the predicted values of each dataset [36]. A *p*-value < 0.05 was considered significant for all the analyses, except for the multiple comparisons, for which Bonferroni correction was used (*p*-value per number of contrast in the five groups: *p*-value < 0.01). The analyses were performed using the STATA software (Stata Corp. 2013. STATA Statistical Software, release13, Stata Corp LP, College Station, TX, USA).

### 2.4. Map of Priority Surveillance Areas

The risk profiles for each grid were generated based on values obtained from the MELR models, and the probability of the event (outcome 1 or 2) for each grid was computed and saved as one of several new variables: *p1tr* (predictor of outcome 1 in the training dataset), *p2tr* (predictor of outcome 2 in the training dataset), *p1ts* (predictor of outcome 1 in the test dataset), and *p2ts* (predictor of outcome 2 in the test dataset). The predictor probability was calculated through the logit function (Equation (3)), which is able to transform the unit interval [0, 1] into a straight line [+∞, −∞]:(3)$$p\left(y\right)=\frac{1}{1+e^{-\beta_{0}+\sum \beta_{n}X}}.$$

The new variables are the result of three different layers. The first layer describes the probability to positively detect a virus, and the second describes the probability to positively detect a young seropositive animal. A third layer was created for the amount of compliance. The kernel smoothing function [37] implemented in ArcGIS was used to distribute the estimated risk throughout the entire area. The bandwidth for kernel regression estimation was obtained by maximizing the cross-validation log-likelihood function [38]. The raster dataset was also converted to its polygon features. The final risk map aimed at identifying the areas with the highest probability to find pockets of infection was generated by the overlapping spaces of the three layers.

## 3. Results

The training dataset (Goceano–Baronia area) consisted of 3953 grids, while the test dataset (Barbagia–Ogliastra area) included 1896 grids with similar altitude (mean = 618, SD = 253), length of roads (km, median = 0.9, IQR = 0–2.3), and amount of protected forest (km^2^, median = 0, IQR = 0–1). The Goceano–Baronia and Barbagia–Ogliastra areas differed for the amount of forest (km^2^, training dataset median = 1.1, IQR = 0.6–1.9; test dataset median = 2.2, IQR = 1.3–2.5), wild boar density estimation (training dataset median = 1, IQR = 0–5; test dataset median = 3, IQR = 0–6) [13], and the absence of free-ranging pigs in the first but present in the second area. Table 1 summarizes the features related to ASF in wild boar in both the Goceano–Baronia and the Barbagia–Ogliastra areas: number of wild boar tested, PCR-positive, seropositive, and the compliance value. Data from 6488 wild boar tested within the Goceano–Baronia were used to fit the two MELR models.

Outcome 1 (PCR-positive wild boar in 2018–2019) and outcome 2 (seropositive young animals in 2019–2020) are presented in Figure 3 as the positive grids observed and the neighboring grids imputed, considering the average distance as 5 km^2^ for the radius [31]. The descriptive statistics in Table 2a,b show the baseline variable distribution in the training dataset (Gogeano–Baronia) based on both outcomes. Given the values imputation in the neighboring grids, over the 3953 grids of the Goceano–Baronia dataset, 178 presented the outcome 1 (presence of PCR-positive), and 229 the outcome 2 (presence of young seropositive).

### 3.1. Mixed-Effects Logistic Regression Model Results

The final results of the MELR models are reported in Table 3a,b.

The contribution of both PCR-positive and seropositive young and adult were evaluated in both models by a stepwise process. The first model aimed at estimating the main factors contributing to the detection of PCR-positive. The final model indicates a relevant and statistically significant effect of PCR-positives (dichotomous variable 0/1) detected the previous year (OR_adj_ = 18.71, 95% IC = 11.55–30.29, *p*-value < 0.0001). Each adult seropositive animal detected during the same hunting season increased the probability of finding a PCR-positive (OR_adj_ = 1.83, 95% IC = 1.66–2.01, *p*-value < 0.0001). A not statistically significant coefficient was estimated for young seropositive wild boar. The grids located at altitude over 500 mamsl showed about seven times more the probability to find virus respect to those at altitude ≤ 500 mamsl (OR_adj_ = 7.691, 95% CI = 4.978–11.881, *p*-value < 0.0001).

Considering that forest and wild boar density were highly correlated (Spearman correlation coeff. = 0.7636, *p*-value < 0.0001), only the forest was included in the final MELR model, thereby increasing the R^2^ value by 0.12 points, decreasing the AIC by 5.5 points, and decreasing the BIC by 3.2 points, compared to animal density. The probability of observing a PCR-positive animal increased by about 1.5 times for every 1 km^2^ of forest (OR = 1.53, 95% CI = 1.18–2.52, *p*-value = 0.001). The final AIC and BIC values were 401.6 and 487.2, respectively. Random intercepts in the output exhibited significant variation based on a likelihood-ratio test versus a one-level binomial regression model (coeff. = 3.71, *p*-value = 0.025) and the SD of random intercepts (3.128) was greater than twice the standard error (SE, 0.905). This result favors the random-intercept model, indicating that there is significant variation in the number of PCR-positive results between grids. Conditional on virus-positive and seropositive results, altitude, and animal density, we estimated that the latent responses within the same grids had a large correlation of approximately 0.78. Thus, 78% of the variance of a latent response was explained by the between-grid variability. The predicted values generated by this model constituted the “*virus layer*”.

The second MELR model used the young seropositive wild boar detected in 2019–2020 as the outcome; employed the adult seropositive animals, wild boar density, amount of forest, and altitude higher/lower than 500 mamsl as the fixed effects; and applied a random effect for each grid. Increasing by 1 the number of seropositive adult wild boar in the same grid during the same year increased the probability of finding a young seropositive animal by two times (OR_adj_ = 2.07, 95% CI = 1.53–2.80, *p*-value < 0.0001). The probability to observe the outcome was increased by 4% (OR_adj_ = 1.04, 95% CI = 1.01–1.07, *p*-value = 0.01) by the increase of the animal density in the grid by one wild boar. In grids located over 500 mamsl, the probability to detect young seropositive animal was about 1.5 times more with respect to those located under 500 mamsl (OR_adj_ = 1.68, 95% CI = 1.27–2.22, *p*-value < 0.0001). The probability of observing a PCR-positive animal increased by about two times for every 1 km^2^ of forest (OR = 2.13, 95% CI = 1.77–2.54, *p*-value < 0.0001). The final AIC and BIC values were 462.1 and 499.6, respectively. The better fit of random-intercept compared to the simple logistic model was demonstrated by the likelihood-ratio test (coeff. 19.01, *p*-value = 0.002) and the random intercept’s SD and SE values (1.145, 0.699), considering that 90% of the variance of a latent response could be explained by the between-grid variability (ICC = 0.90). The predicted values generated by this model constitute the *“young seropositive layer”.*

In both models, the amount of road did not show any statistically significant effect. The explorative multivariable analyses based on the stepwise variable selection highlighted the controversial role of the amount of protected forest (by 1 km^2^). This variable had an apparently protective role in the probability of finding a PCR-positive animal (OR = 0.97, 95% CI = 0.96–0.99, *p*-value = 0.001) but increased the probability of detecting a young seropositive animal (OR = 1.01, 95% CI = 1.005–1.01, *p*-value < 0.0001) in the second model.

Considering that areas where hunting is not practiced corresponded to zero compliance, this variable was excluded in both the MELR models and included as areas with a lack of information in the “*compliance layer*”. The final map shown in Figure 4 represents the interpolation between all informative layers and indicates the target areas where the need for active carcass research is a priority.

### 3.2. Model Validation

Using the model’s regression equation, the predicted probability of a positive event was computed for the grids in both the datasets and for each outcome. Values of 0.365 and 0.483 (the medians of the predicted values) were selected as cut-off points for outcomes 1 and 2, respectively, and a grid was declared positive if its predicted event probability was larger than its cut-off point. Internal and external validation were performed by evaluating both the predicted values in training and test datasets for outcome 1 and outcome 2. Table 4 shows the contingency tables between the observed and predicted values, while Figure 5 shows the distribution of the C statistics.

In the training dataset, the model based on outcome 1 showed a sensitivity (the ability to correctly classify the positive grids) of 94.4% and a specificity (the ability to correctly identify negative grids) of 99.8%. The fitting distribution of the C statistic (0.913) indicated excellent discrimination with a median of 0.919, and the distribution was concentrated around the median (QRNG/2 = 0.032; C Var = 3%). Furthermore, the minimum value of the distribution was 0.861, which is still an acceptable value for discrimination.

The model fitted to the training dataset based on outcome 2 was characterized by 96.9% sensitivity and specificity of 99.2%. The C statistic calculated for the 3953 grids was equal to 0.884, with a median of 0.889 (QRNG/2 = 0.037; C Var = 4%) and a minimum value of 0.836, indicating the good fit of the model in discriminating between the two classes of outcomes.

Similar values of sensitivity and specificity were obtained for the test dataset in both outcome 1 (sensitivity = 92.9%, sensibility = 94.9%) and outcome 2 (sensitivity = 96.4%, sensibility = 97.0%). The validation distributions of the C statistics for outcomes 1 and 2 in the test dataset were not as good as the previous results but still proved to offer acceptable discrimination of the model for grid validation. The median values of the distribution were 0.725 (QRNG/2 = 0.062; C Var = 8.5%) and 0.718 (QRNG/2 = 0.085; C Var = 11.8%), and nearly 25% of the distribution was larger in value than 0.8. Thus, we can conclude that the model perfectly discriminated the original sample and also accurately discriminated outside of the sample

## 4. Discussion

The fundamental role of well-organized active and passive surveillance in the fight against ASF was underlined by the European Commission working document SANTE/7113/2015 [23]. However, one of the drawbacks of passive surveillance is the large amount of land that must be checked, particularly in the last phase of eradication when viral evidence is very low. Otherwise, passive surveillance based on active research of carcasses is notoriously expensive in terms of time and costs. The present work aimed at defining a standardized methodology able to identify relevant areas to search for wild boar carcasses or other samples based on a scale of priorities.

As indicated by the data, ASF in Sardinia is limited to the areas of Goceano–Baronia and Barbagia–Ogliastra, which are included in the wild boar infected zone in the middle of the region. The small amount of data from passive surveillance and the data from active surveillance being limited to November–January might be an impediment for the final Sardinian ASF Exit Strategy [22]. Furthermore, the animals tested out of hunting season, which are almost limited to those killed by car accidents, are not sufficient to establish the disease’s lethality. Thus, a well-planned study of carcasses limited to specific areas aimed to detect possible virus reservoirs is fundamental in Sardinia in the last phases of eradication.

The most recent carcass research dates back to April 2019, and corresponds with the last detection of the virus. At the end of the hunting season in 2018–2019, within the bounds of a protected forest in Bultei municipality, evident viral circulation was detected (2 virus-positives and 2 young seropositives). In this forest, hunting is forbidden, leading to an absence of information. Two PCR-positive carcasses were found during carcass research activities inside this area. This forest represents an example area, where the probability of finding an eventual pocket of infection is strictly related to a lack of information, presence of PCR-positive, and young seropositive animals. These findings prompted us to develop a standard procedure to be carried out over the entire infected area.

The MELR models applied in the area of Goceano–Baronia accurately describe the probability of detecting both PCR-positive and young seropositive animals with sensitivity and specificity higher than 94%. These two outcomes in combination with a lack of information (low compliance) were confirmed to be the main indicators of conceivable disease persistence. External and internal validation were also performed, providing excellent results. The regression equation underlying these models has a discrimination property that is not specific to the available sample but is extendable to other populations, since it is able to accurately discriminate data that are new and independent from the sample used to fit the model. Thus, this procedure could be applied in other European countries close to ASF eradication to identify the areas where further surveillance is needed. Furthermore, not only passive surveillance but also active surveillance (e.g., a selective hunt) could be programmed based on this work, if necessary.

## 5. Conclusions

This work provides a fundamental instrument for authorities and stakeholders in decision analysis. A probability map and intervention priority scale represent the first steps for well-programmed surveillance. Moreover, passive surveillance is based on field activities; therefore, an accurate program should not be based only on epidemiological and statistical analyses but also on the available resources. The importance of training both hunters and staff employed in the active search for wild boar carcasses was previously underlined [23]. Furthermore, as demonstrated by the same authors, good collaboration between hunters and veterinary services can lead to greater efficacy in ASF detection [39].

## Figures and Tables

**Figure 1 vaccines-08-00723-f001:**
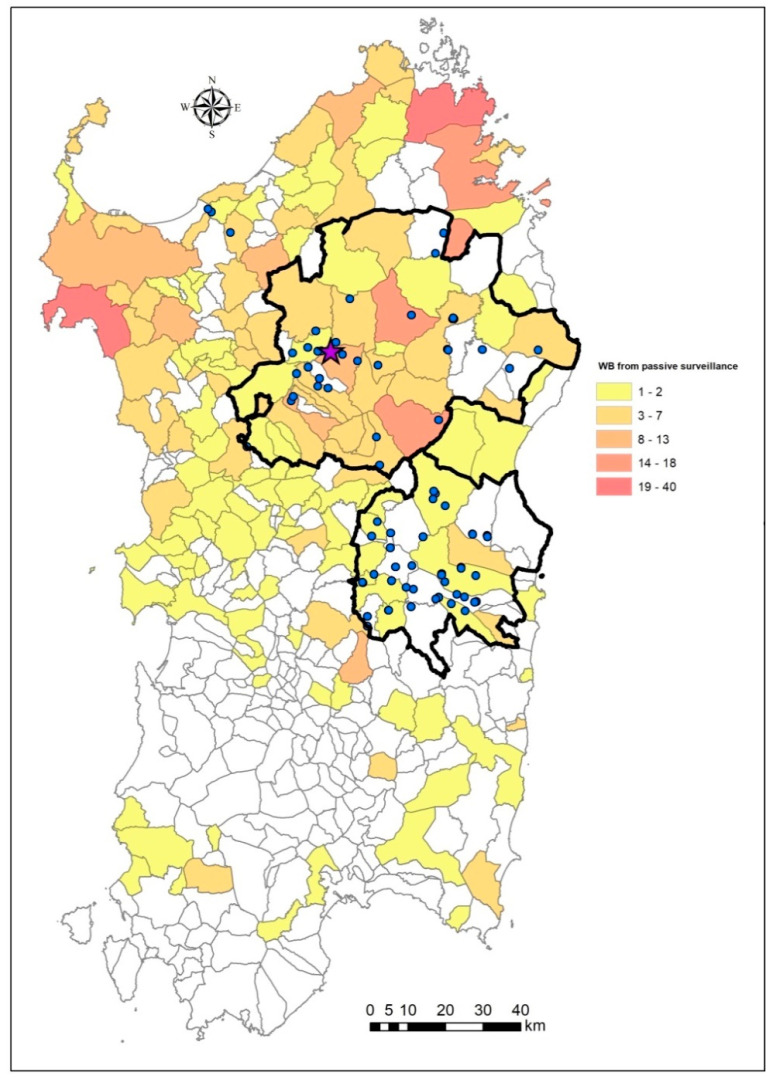
Epidemiological context of African swine fever in Sardinia from April 2019, and updated on October 2020. The violet star indicates the last two PCR-positive wild boar detected in April 2019, in the protected forest of Bultei (Sassari). The blue dots indicate the seropositive wild boar (WB). The proportion of animals tested via passive surveillance in each municipality is presented as a cloropletic map. The bold black lines define the limits of the two areas of Goceano–Baronia (the largest) and Barbagia–Ogliastra (the smallest).

**Figure 2 vaccines-08-00723-f002:**
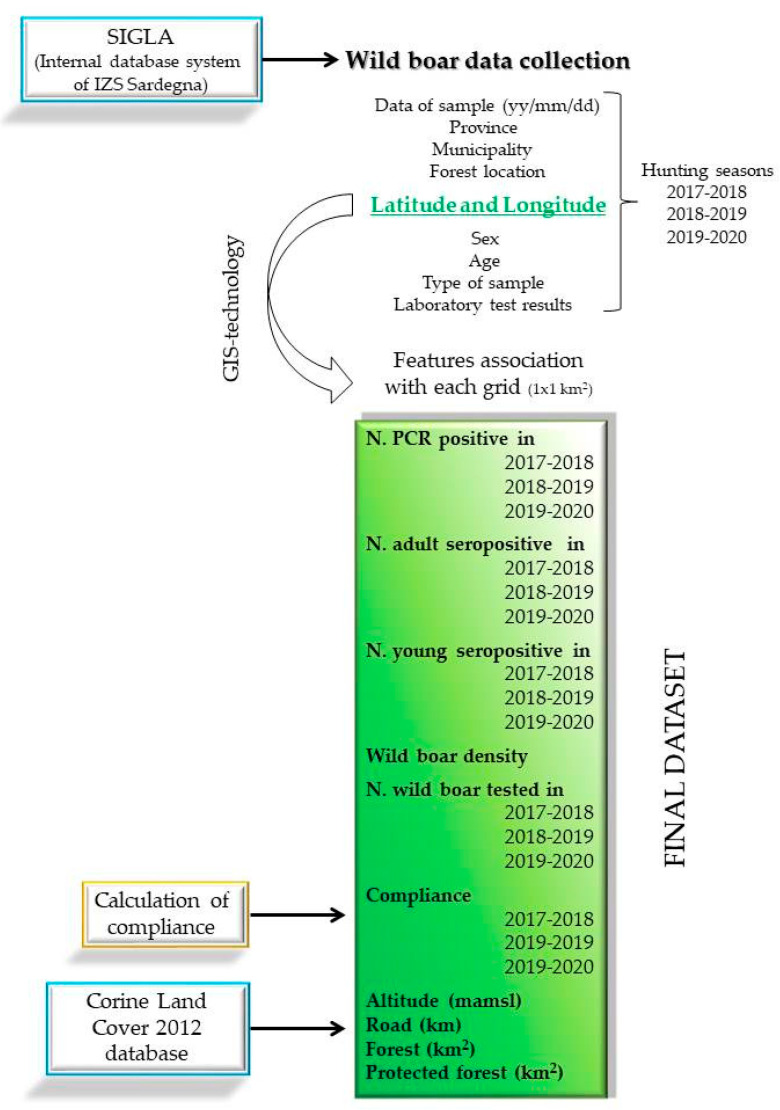
Flow-chart representing the study design methodology. Wild boar data were collected from the Internal database system (SIGLA) of Istituto Zooprofilattico Sperimentale della Sardegna. Based on latitude and longitude (green font), variables were associated with each grid of 1 km^2^ by year, and the final dataset (green square) was implemented with the calculated compliance and the land features (altitude, road, forest, protected forest) collected by Corine Land Cover 2012 database.

**Figure 3 vaccines-08-00723-f003:**
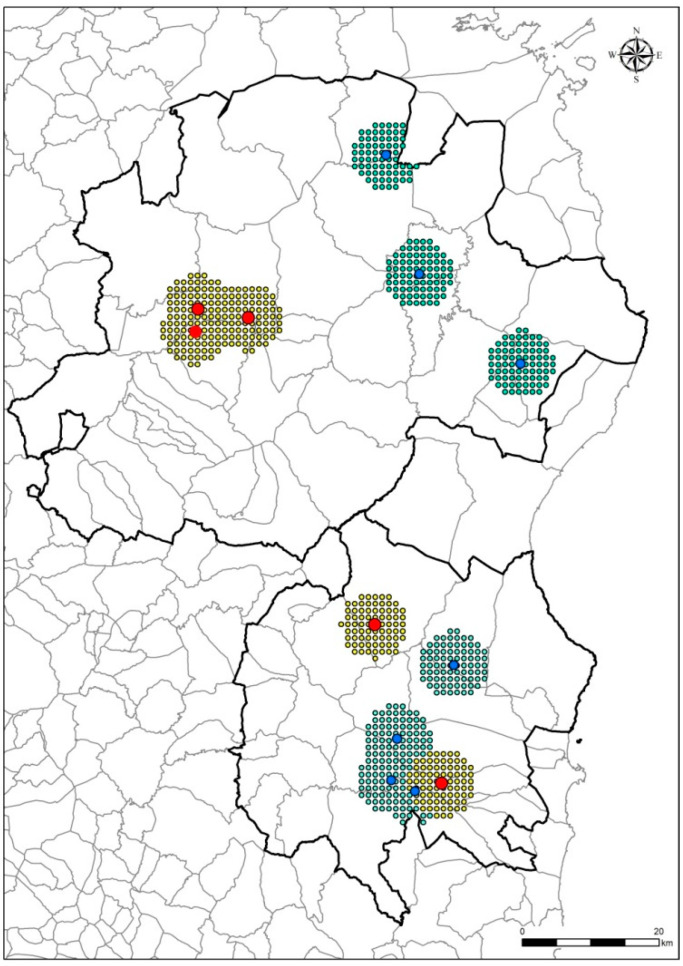
African swine fever PCR-positive animals detected in 2018–2019 (outcome 2—red dots) and seropositive young animals in 2019–2020 (outcome 2—blue dots) in the two areas of Goceano–Baronia and Barbagia–Ogliastra (bold black lines). Points around the red and blue dots identify that the neighboring grids were positive values were imputed, considering the 5 km^2^ for the radius.

**Figure 4 vaccines-08-00723-f004:**
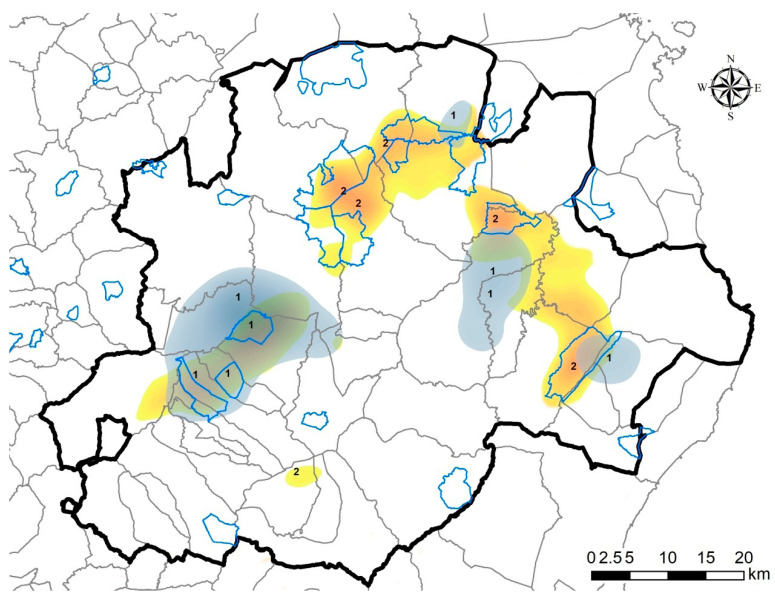
Final passive surveillance map. The grey areas are areas where the probability of finding a pocket of infection is higher (areas 1) given the mixed-effect logistic regression (MELR) model results and the lack of information. The areas indicated as number 2 are areas where the lack of information is higher but not indicated by the MELR models. The blue lines indicate the limits of the protected forest where hunting is forbidden.

**Figure 5 vaccines-08-00723-f005:**
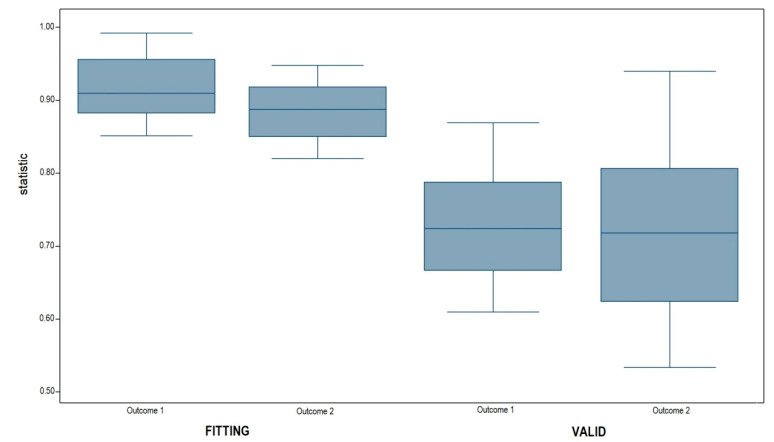
Discrimination of the models: fitting values versus validation distribution for the four MELR models. Distributions are represented as box-plots: the line splitting of each box in two represents the median value; the bottom and top edges of each box represent the lower and upper quartiles; the values at which the vertical lines stop are the upper and lower values of the data.

**Table 1 vaccines-08-00723-t001:** Descriptive analyses of the Goceano–Baronia and Barbagia–Ogliastra areas on the data included in the models. Data are presented as the frequency (prevalence expressed as %) and median (interquartile range).

Area	Goceano–Baronia (3953 km^2^)				Barbagia–Ogliastra (1896 km^2^)			
Hunting Season	Wild Boar Tested	PCR-Positive	Seropositive	Compliance	Wild Boar Tested	PCR-Positive	Seropositive	Compliance
2017–2018	2104	6 (0.28%)	106 (5.03%)	30.7 (18.4–42.7)	497	10 (2.01%)	61 (12.27%)	14.5 (7.0–27.6)
2018–2019	2175	4 (0.18%) ^1^	39 (1.79%)	28.6 (15.9–53.9)	654	2 (0.31%)	48 (7.34%)	17.9 (10.4–36.0)
2019–2020	2209	0 (0%)	32 (1.45%)	32.5 (19.7–48.2)	702	0 (0%)	43 (6.12%)	18.6 (11.0–25.5)
Total	6488	8 (0.12%)	177 (2.72%)	31.8 (18.3–50.6)	1853	12 (0.65%)	152 (8.20%)	17.6 (10.1–28.5)

^1^ Two of these samples arise from passive surveillance activities in April 2019 in Bultei municipality.

**Table 2 vaccines-08-00723-t002:** Descriptive statistics of variables in the training dataset (Goceano–Baronia) for (a) outcome 1 (PCR-positive wild boar detected in 2018–2019) and (b) for outcome 2 (young seropositive animals detected in 2019–2020). Data are presented as the frequency (%) or mean (standard deviation (SD)), median (I–III interquartile), and *p*-value.

(a) Variables	Outcome 1 = 1 PCR-Positive Detected in 2018–2019 (178 Grids)	Outcome 1 = 0 PCR-Positive Not Detected in 2018–2019 (3775 Grids)	*p*-Value
PCR-positive ^1^			
Hunting season 2017–2018	1 (0–2)	0 (0–0)	<0.0001
Adult seropositive ^1^			
Hunting season 2017–2018	6 (4–8)	0 (0–2)	<0.0001
Hunting season 2018–2019	3 (2–9)	0 (0–1)	<0.0001
Young seropositive ^1^			
Hunting season 2017–2018	2 (1–3)	0 (0–0)	<0.0001
Hunting season 2018–2019	0 (0–0)	0 (0–0)	NS
Altitude (mamsl)	700 (600–800)	500 (300–700)	<0.0001
Road (km)	1.7 (0–2.6)	1.0 (0–2.4)	0.007
Forest (km^2^)	1.82 (0.81)	1.24 (0.82)	<0.0001
Wild boar density ^1,2^	5 (0–7)	0 (0–5)	<0.0001
Amount of protected forest (km^2^)	0 (0–1.3)	0 (0–0.7)	<0.0001
**(b) Variables**	**Outcome 2 = 1** **Young Seropositive Animal Detected in 2019–2020** **(229 Grids)**	**Outcome 2 = 0** **Young Seropositive Animal** **Not Detected in 2019–2020** **(3724 Grids)**	***p*-Value**
PCR-positive ^1^			
Hunting season 2017–2018	0 (0–0)	0 (0–0)	NS
Hunting season 2018–2019	0 (0–0)	0 (0–0)	NS
Adult seropositive ^1^			
Hunting season 2017–2018	1 (0–4)	0 (0–2)	<0.0001
Hunting season 2018–2019	0 (0–1)	0 (0–1)	NS
Hunting season 2019–2020	0 (0–2)	0 (0–1)	<0.0001
Young seropositive ^1^			
Hunting season 2017–2018	0 (0–0)	0 (0–1)	NS
Hunting season 2018–2019	0 (0–0)	0 (0–0)	NS
Altitude (mamsl)	550 (450–700)	500 (300–700)	0.0026
Road (km)	0 (0–2.18)	1.1 (0–2.41)	<0.0001
Forest (km^2^)	1.78 (0.69)	1.24 (0.82)	<0.0001
Wild boar density ^1,2^	3 (0–5)	0 (0–5)	<0.0001
Amount of protected forest (km^2^)	1.58 (0–4.2)	0 (0–0.05)	<0.0001

^1^ Considering the features of wild boar imputed in the neighboring grids, the number of positive animals correspond to the number of grids covered by a 5 km^2^, for which the ASF-positive cases were imputed. ^2^ Wild boar density is based on the wild boar management plan [13].

**Table 3 vaccines-08-00723-t003:** Results of the mixed-effect logistic regression model with (**a**) PCR-positive cases (2018–2019) considering outcome 1 and (**b**) young seropositive cases (2019–2020) considering outcome 2. Data are reported as the adjusted odds ratio (OR_adj_), 95% confidence intervals (95% CI), and *p*-values.

Outcome 1: PCR-Positive 2018–2019	Variables	OR_adj_	95% CI	*p*-Value
	Presence of PCR-positive 2017–2018	18.71	11.55–30.29	<0.0001
	Adult Seropositive 2018–2019	1.83	1.66–2.01	<0.0001
	Altitude >500 mamsl	7.69	4.98–11.88	<0.0001
	Forest (by 1 km^2^)	1.53	1.18–2.52	0.001
		Sd	SE	95% CI
Random-effect	grid	3.128	0.905	1.730–5.652
LR test vs. logistic regression: 3.71, *p*-value = 0.025				
		ICC	SE	95% CI
Residual intraclass correlation	grid	0.782	0.127	0.588–0.919
**Outcome 2:** **Young Seropositive** **2019–2020**	**Variables**	**OR_adj_**	**95% CI**	***p*-Value**
	Adult Seropositive 2019–2020	2.07	1.53–2.80	<0.0001
	Wild boar density	1.04	1.01–1.07	0.028
	Altitude >500 mamsl	1.68	1.27–2.22	<0.0001
	Forest (by 1 km^2^)	2.13	1.77–2.54	<0.0001
		Sd	SE	95% CI
Random-effect	grid	1.145	0.699	0.338–3.717
LR test vs. logistic regression: 19.01, *p*-value = 0.002				
		ICC	SE	95% CI
Residual intraclass correlation	grid	0.906	0.002	0.893–0.998

**Table 4 vaccines-08-00723-t004:** Contingency tables for the discriminatory tests applied for internal validation (outcome 1 and outcome 2 on the training dataset), and eternal validation (outcome 1 and outcome 2 on the test dataset).

Model Outcome		*Outcome 1*	*Outcome 2*
Dataset		Observed
Training dataset			1	0	tot	1	0	tot
**Predicted**	1	168	6	174	222	28	250
0	10	3769	3779	7	3696	3703
tot	178	3775	3953	229	3724	3953
Sensitivity	94.4%	96.9%
Specificity	99.8%	99.2%
Test dataset			1	0	tot	1	0	tot
**Predicted**	1	143	89	232	242	50	292
0	11	1653	1664	9	1595	1604
tot	154	1742	1896	251	1645	1896
Sensitivity	92.9%				96.4%		
Specificity	94.9%				97.0%

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
