# Peer review of "Standardized Methodology for Target Surveillance against African Swine Fever"

_vaccines, 2020, doi:10.3390/vaccines8040723_

Round 1

Reviewer 1 Report

The need for cost-effective but thorough surveillance for ASF in wild boars is undeniable and becomes particularly important as a country or territory moves towards declaring that ASF has been eradicated based on absence of circulation of the virus. The details of the methodology are outside my fields fo expertise. However, the broad range of data incorporated is impressive and the provided results that suggest the conclusions are likely to be valid.

The English needs considerable attention, preferably from a first language English speaker. It is mostly more or less possible follow but has many mistakes and an un-English style, e.g. in English we would not use ‘The surveillance’ or ‘the passive surveillance’, just ‘surveillance’ and ‘passive surveillance’. In some places it does become difficult to follow.

Specific comments:

Line 18: Perhaps ‘pig disease’ rather than ‘animal disease’, of course I agree that ASF is the most important animal disease, but probably not everybody will.

Line 42: spp should not be italicised, and replace ‘insect’ with ‘arthropod’, as ticks are not insects (one pair of legs too many). 

Table 1 is placed before the basis of the ‘Training’ dataset and the ‘Test’ dataset has been explained. This is rather confusing for the reader so perhaps the explanation of these terms, provided in line 192 near the bottom of the next page, could be provided earlier. Alternatively the table could possibly be moved to after the description, although it is currently placed where it is first referred to.

Lines 148-149: The sentence needs rewriting as the meaning is not clear, I think due to language difficulties. Would ‘joins’ perhaps mean ‘links’?

Line 320: Should Figure 6 be Figure 5? There are only five figures but I am not sure why there is no map of the test dataset results.

Table 3: The caption mentions both the training and test datasets, but only the training dataset in indicated in the table itself.

Author Response

The need for cost-effective but thorough surveillance for ASF in wild boars is undeniable and becomes particularly important as a country or territory moves towards declaring that ASF has been eradicated based on absence of circulation of the virus. The details of the methodology are outside my fields of expertise. However, the broad range of data incorporated is impressive and the provided results that suggest the conclusions are likely to be valid.

Dear reviewer,

In the name of all the co-authors we would like to really thank you for this important and fast revision of this manuscript. Furthermore, we would thank you for the understanding of the importance of this work in the last phase of the eradication process, aimed to detect the last possible or eventually residual of ASF virus. The presented procedure could be a valid tool to save the resources and use these in the best way during surveillance activities.

The English needs considerable attention, preferably from a first language English speaker. It is mostly more or less possible follow but has many mistakes and an un-English style, e.g. in English we would not use ‘The surveillance’ or ‘the passive surveillance’, just ‘surveillance’ and ‘passive surveillance’. In some places it does become difficult to follow.

Thank for this suggestion. English is not our first language so the MDPI English revision service has been used to completely review the language in mother tongue speaking.

Specific comments:

Line 18: Perhaps ‘pig disease’ rather than ‘animal disease’, of course I agree that ASF is the most important animal disease, but probably not everybody will.

We agree with this suggestion and animal disease has been substituted with pig disease

Line 42: spp should not be italicised, and replace ‘insect’ with ‘arthropod’, as ticks are not insects (one pair of legs too many). 

Dear reviewer, we are really sorry and ashamed for this mistake… it was certainly an oversight! Of course, insect has been substituted with arthropod

Table 1 is placed before the basis of the ‘Training’ dataset and the ‘Test’ dataset has been explained. This is rather confusing for the reader so perhaps the explanation of these terms, provided in line 192 near the bottom of the next page, could be provided earlier. Alternatively the table could possibly be moved to after the description, although it is currently placed where it is first referred to.

Considering your right suggestion, we have rename the caption of Table 1 as the two areas (Goceano-Baronia and Barbagia-Ogliastra).Following the suggestion of reviewer 4, we completely reviewed the table and moved it in Result section (after the definition of the use of these two datasets). Many thanks for your help.

Lines 148-149: The sentence needs rewriting as the meaning is not clear, I think due to language difficulties. Would ‘joins’ perhaps mean ‘links’?

The entire phrase has been review by the language service as: This procedure has been realized with the use of ArcGIS (ArcMap 10.4) applying the tool “joint” to link the layers build for each variable included.

Line 320: Should Figure 6 be Figure 5? There are only five figures but I am not sure why there is no map of the test dataset results.

Dear review, it was a mistake. The figures are five, and we have not produced another figure to not weight the paper with a lot of figures, but if you think that is better to have this one we can produce it. Thank’s for this reporting.

Table 3: The caption mentions both the training and test datasets, but only the training dataset in indicated in the table itself.

Dear review, it was a mistake. We have added the value observed and predicted for the test dataset. Thank you for this reporting.

Reviewer 2 Report

Cappai and co-workers write a nice and interesting paper. However, I think  the  paper is not in line with the aims and topic of Vaccines.

Author Response

Cappai and co-workers write a nice and interesting paper. However, I think  the  paper is not in line with the aims and topic of Vaccines.

Dear reviewer,

Thank you very much for the revision of our manuscript and for the interesting in this lecture. The paper has been submitted to a Special Issue “African Swine Fever Virus Prevention and Control”. The call of this SI’s Editor said “[…] Due to the severity of the current African swine fever virus epidemic, alternative strategies including prophylaxis may also need to be considered. Deployment of such novel control strategies will require the development of effective implementation strategies based on robust epidemiological modelling”. We believe that this paper could provide an interesting novel control strategy, based on epidemiological model and, for this reason, we choose this Journal.

Reviewer 3 Report

This is an interesting MS that i am not sure fits fully Vaccine but more with Viruses. Nonetheless, the MS uses an interesting approach to address ASFV in Europe. 

The MS really need a lot of help with language. I have tried my best to help improve this MS (see attached edited pdf document). I also recommend that the authors move away from the use of the word cell, rather use grid (i have edited this thought the MS). 

Author Response

This is an interesting MS that i am not sure fits fully Vaccine but more with Viruses. Nonetheless, the MS uses an interesting approach to address ASFV in Europe. 

Dear reviewer,

Thank you very much for the revision of our manuscript and for the precious suggestions. The paper has been submitted to a Special Issue “African Swine Fever Virus Prevention and Control”. The call of this SI’s Editor said “[…] Due to the severity of the current African swine fever virus epidemic, alternative strategies including prophylaxis may also need to be considered. Deployment of such novel control strategies will require the development of effective implementation strategies based on robust epidemiological modelling”. We believe that this paper could provide an interesting novel control strategy, based on epidemiological model and, for this reason, we choose this Journal.

The MS really need a lot of help with language. I have tried my best to help improve this MS (see attached edited pdf document). I also recommend that the authors move away from the use of the word cell, rather use grid (i have edited this thought the MS).

Thank for this suggestion. English is not our first language so the MDPI English revision service has been used to completely review the language in mother tongue speaking. Furthermore, we have change the manuscript following your very kind correction within all the paper (grid instead of cell, and so on).

Reviewer 4 Report

The work is interesting and aims to identify the higher risk areas for the persistence of the ASFV based on epidemiological models.

Minor coments:

Line 56-66 In the introduction there seems to be an exchange of concepts between active and passive surveillance. Please check this out.

Line 117 In the legend of figure 1 wouldn't it be April, 2019?

Line 119 Remove the second parenthesis.

Lines 133-138 I find this information unnecessary.

Table 1: What is the purpose of separating "PCR positive" from "Young PCR positive" and "Seropositive" from "Young seropositive"? This table is very confusing. I suggest you remove the variables altitude, road, forest, wild boar density ... and leave this information in the text. Modify the table by placing the evaluated periods in the first column and in the other columns the results of the molecular and serological tests. I suggest that the other tables are also modified.

Major coments:

The methodology is confused. It is not clear the number of animals evaluated in the models. I suggest making a flow chart to make the study design easier to understand. Why was a model not proposed with data from one period and validated with information from another period?

It is not clear why the outcome and different periods are used in the models [The outcome 1 (virus positive wild boar in 2017-2018) and outcome 2 (seropositive Young animals in 2019-2020)]. In addition, it seems strange to use different intervals in the construction of the mixed-effect logistic regression model than the one proposed in the other model (Table 3).

The results presented in tables 3 and 4 show significant variables with confidence intervals very close to 1, which can be questionable. Table 3: Altitude (m) 1,004 (CI95% 1,002-1,005) and in Table 4 Wild boar tested 2019-2020 OR 0.93 (CI95% 0.87-0.99) and Altitude (m) OR1,001 (CI95% 1,0006-1,002).

Author Response

The work is interesting and aims to identify the higher risk areas for the persistence of the ASFV based on epidemiological models.

Minor coments:

Line 56-66 In the introduction there seems to be an exchange of concepts between active and passive surveillance. Please check this out.

Dear review, in the working document 7113/2015 the passive surveillance is defined as the “active” search of carcasses. These terms could generate misunderstanding, as you underline, so the word “active” referred to passive surveillance has been deleted and the paragraph has been reviewed by English translation service.

Line 117 In the legend of figure 1 wouldn't it be April, 2019?

Yes, thank you very much for reporting this mistake.

Line 119 Remove the second parenthesis.

Yes, thank you very much for reporting this mistake.

Lines 133-138 I find this information unnecessary.

The authors agree with you and the line 133-138 have been deleted as you suggested. Thank you

Table 1: What is the purpose of separating "PCR positive" from "Young PCR positive" and "Seropositive" from "Young seropositive"? This table is very confusing. I suggest you remove the variables altitude, road, forest, wild boar density ... and leave this information in the text. Modify the table by placing the evaluated periods in the first column and in the other columns the results of the molecular and serological tests. I suggest that the other tables are also modified.

Dear review,

The overall PCR and seropositive have been separated by young given that one of the outcomes is the young seropositive animals. In order to provide a measure of this outcome distribution, this variable was splitted in adult and young. Considering your suggestion the Table 1 has been formatted, the information about land type (altitude, forest, animal density, etc…) have been deleted and included in the text and only overall PCR positive and seropositive are included. Considering the need of p-value test in Table 2a and 2b, and given that each variable has been evaluated for the inclusion in the final regression model on baseline different distribution (expressed in table 2a and 2b), these modification are not feasible for these two tables. Otherwise, to lighten the table, the row of “Seropositive” has been deleted, leaving only the rows about young and adult seropositive animals. Please let me know if agree with this format or you prefer some modifications.

Major coments:
The methodology is confused. It is not clear the number of animals evaluated in the models. I suggest making a flow chart to make the study design easier to understand.

A total of 6488 wild boar hunted within Goceano-Baronia during the hunting season 2017-2020 have been used to fit the two final MELR models. The 1853 wild boar hunted and tested in Barbagia-Ogliastra have been used for external validation. The flow-chart describing the study design has been included as Figure 2, as you suggested.

Why was a model not proposed with data from one period and validated with information from another period?

Dear review, thank you for this very good question. In this work we performed both internal and external validation because the hunt could be differently managed between two or three years and some bias could be included. Furthermore, considering that this procedure would be valid not only for Sardinia but even for other countries, only an internal validation would not have guaranteed its applicability. This methodology has been specified in line 185: “Data from Goceano-Baronia has been used as “Training dataset” to fit the models while Barbagia-Ogliastra database was used as “Test dataset” for external validation. Specification for internal validation has been included in line “Internal validation has been performed using two different period of data: 2018-2019 vs 2017-2018 to outcome 1, and 2019-2020 vs 2018-2019 to outcome 2.”

It is not clear why the outcome and different periods are used in the models [The outcome 1 (virus positive wild boar in 2017-2018) and outcome 2 (seropositive Young animals in 2019-2020)]. In addition, it seems strange to use different intervals in the construction of the mixed-effect logistic regression model than the one proposed in the other model (Table 3).

Dear review, in the second paragraph of statistical analysis the authors explain the main outcome of the models: “Considering the previous risk analyses performed by the same authors [11,30], the main risk factors indicating the presence of not detected pocket of infection are the more recent PCR positive, the presence of young seropositive animals or the absence of information”

As you can see in table 1, no virus positive has been detected during the last hunting season (2019-2020). Otherwise, some young seropositive animals still be found in this hunting season. To give a more recent as possible results, we used these two different period, considering that the models are separate each other and these are used to associate the outcome probability at each variable included. To better explain this concept we include in line 162 “To provide more recent results, given the completely absence of PCR positive during the last hunting season, but the persistence of young seropositive animals, the outcome 1 and 2 have been associated to two different periods: hunting season 2018-2019 and hunting season 2019-2020, respectively.”

The results presented in tables 3 and 4 show significant variables with confidence intervals very close to 1, which can be questionable. Table 3: Altitude (m) 1,004 (CI95% 1,002-1,005) and in Table 4 Wild boar tested 2019-2020 OR 0.93 (CI95% 0.87-0.99) and Altitude (m) OR1,001 (CI95% 1,0006-1,002).

Dear review, thank you for this reporting. Consider that the probability to find a virus positive or young seropositive increase of 0.4% by each metre of altitude more. This measure could seems to be low and to near to the value one, but if you relate this value to 50 or 100 metres of altitude the odds ratio increases significantly. The same for the odds increase by each wild boar in 1x1 km2 grid. We have translate this value and report in the table as attitude > 500 mamsl, but considering the aim of this work (produce a spatial model by 1x1 km grid) it is not advisable. Anyway, considering the need of point spatial risk projection, this categorization will not be used for the final map but the predictor of altimetry by 1 metres of mamsl. But please, let me know if you prefer this form.

Round 2

Reviewer 4 Report

The author made the suggested changes and clarified the doubts of the methodology and clarified them. Thus, the quality of the manuscript has been improved in this revised version.